# siRNA Nanoparticle Targeting PD-L1 Activates Tumor Immunity and Abrogates Pancreatic Cancer Growth in Humanized Preclinical Model

**DOI:** 10.3390/cells10102734

**Published:** 2021-10-13

**Authors:** Jae Yun Jung, Hyun Jin Ryu, Seung-Hwan Lee, Dong-Young Kim, Myung Ji Kim, Eun Ji Lee, Yeon-Mi Ryu, Sang-Yeob Kim, Kyu-Pyo Kim, Eun Young Choi, Hyung Jun Ahn, Suhwan Chang

**Affiliations:** 1Department of Biomedical Sciences, University of Ulsan College of Medicine, Seoul 05505, Korea; flandus@naver.com (J.Y.J.); mshj8018@naver.com (H.J.R.); shlee.daniel@gmail.com (S.-H.L.); dyk850@gmail.com (D.-Y.K.); mj7944@naver.com (M.J.K.); hyloveej6@hanmail.net (E.J.L.); 2Asan Medical Center, Asan Institute for Life Sciences, Seoul 05505, Korea; nordica62@gmail.com (Y.-M.R.); sykimbear@gmail.com (S.-Y.K.); 3Asan Medical Center, Department of Oncology, Seoul 05505, Korea; kkp1122@amc.seoul.kr; 4Center for Theragnosis, Biomedical Research Institute, Korea Institute of Science and Technology, Seoul 05505, Korea; 5Department of Physiology, University of Ulsan College of Medicine, Seoul 05505, Korea

**Keywords:** siRNA, nanoparticle, pancreatic cancer, PD-L1, immunotherapy, humanized NSG

## Abstract

Pancreatic cancer is characterized by late detection, frequent drug resistance, and a highly metastatic nature, leading to poor prognosis. Antibody-based immunotherapy showed limited success for pancreatic cancer, partly owing to the low delivery rate of the drug into the tumor. Herein, we describe a poly(lactic-co-glycolic acid;PLGA)-based siRNA nanoparticle targeting PD-L1 (siPD-L1@PLGA). The siPD-L1@PLGA exhibited efficient knockdown of PD-L1 in cancer cells, without affecting the cell viability up to 6 mg/mL. Further, 99.2% of PDAC cells uptake the nanoparticle and successfully blocked the IFN-gamma-mediated PD-L1 induction. Consistently, the siPD-L1@PLGA sensitized cancer cells to antigen-specific immune cells, as exemplified by Ovalbumin-targeting T cells. To evaluate its efficacy in vivo, we adopted a pancreatic PDX model in humanized mice, generated by grafting CD34^+^ hematopoeitic stem cells onto NSG mice. The siPD-L1@PLGA significantly suppressed pancreatic tumor growth in this model with upregulated IFN-gamma positive CD8 T cells, leading to more apoptotic tumor cells. Multiplex immunofluorescence analysis exhibited comparable immune cell compositions in control and siPD-L1@PLGA-treated tumors. However, we found higher Granzyme B expression in the siPD-L1@PLGA-treated tumors, suggesting higher activity of NK or cytotoxic T cells. Based on these results, we propose the application of siPD-L1@PLGA as an immunotherapeutic agent for pancreatic cancer.

## 1. Introduction

Pancreatic cancer is one of the leading causes of cancer death, mainly owing to the late diagnosis and lack of effective treatment options [1]. Recent clinical trials of immunotherapy for pancreatic cancer exhibited limited success, with only 9% disease control for combined therapy targeting PD-L1 and CTLA4 [2]. Several reasons have been proposed for this disappointing result: (1) the large amount of extracellular matrix surrounding pancreatic cancer cells, including immunosuppressive proteins [3,4]; (2) the recruitment of immune suppressor cells such as Treg and MDSC [5]; (3) the desmoplastic reaction (fibrosis) that prohibits penetration of tumors by therapeutic agents [6]; (4) the low antigenicity of PDAC tumors, which often causes an “immune cold” microenvironment [7].

RNA interference (RNAi) has the ability to suppress oncogenes, tumor suppressor genes, and their regulators [8]. Recently, it has emerged as a promising agent for inducing antitumor immunity in vivo because of its unique advantages, such as the high sequence-specificity for target molecules and “druggable” properties [9]. This strategy is advantageous over antibodies or small molecules, as RNAi-based drugs inhibit the target molecules at the post-transcriptional level rather than at the protein level [10]. Additionally, RNAi-based drugs require delivery of only pico-molar levels of siRNA to tumor cells for suppression of target molecules. In comparison, strategies based on antibodies or small molecules require significantly larger amounts of drugs, such that the molar ratio of the target molecule to the drug is at least 1:1, and may be ineffective if a compensatory expression of target molecules occurs in tumor cells.

PLGA polymers have widely provided efficient drug delivery carriers for chemotherapeutics and nucleotides, due to their low cytotoxicity, biodegradability, sustained-release property, and enhanced permeability and retention (EPR) effect in the medical applications for cancer treatment [11,12,13,14]. Indeed, the Food and Drug Administration has approved several PLGA formulations for drug delivery in humans [15]. Thus, PLGA nanoparticles as siRNA delivery vehicles have drawn great potential in the RNAi-mediated therapeutic applications, in contrast to the commonly used polycationic carriers, which inevitably cause cytotoxic and/or non-degradable issues [11].

Blocking of PD-L1 by silencing is considered a potential strategy for immune checkpoint blockades because such blockades can expose tumor cells to antitumor immunity [16]. For example, a PD-L1 blockade via siRNA-mediated silencing was reported to promote antitumor immunity in immunocompetent mice and suppress melanoma growth [17]. Similarly, a PD-L1 blockade exposed ovarian cancer cells to T-cell killing, leading to significant tumor growth inhibition [18]. Additionally, the recently identified roles of PD-L1 within the intracellular compartments of tumor cells suggest the utility of RNAi-based drugs for blocking immune checkpoints, whereas antibodies do not have access to the intracellular compartments [19]. Pancreatic cancer has proven to be resistant to treatment with immune checkpoint inhibitors such as antibodies [20]. Few studies have adopted the concept of targeting PD-L1 using siRNA for pancreatic cancer. Yoo et al. performed a combined therapy using Gemcitabine plus a PD-L1 siRNA-conjugated magnetic nanocarrier [16]. In a mouse allograft model, this strategy allowed 67% of the animals to survive for 12 weeks, whereas the control animals died after 6 weeks. In another study, the efficacy of combined treatment of a TGF inhibitor and PD-L1 siRNA encapsulated in a pH-responsive clustered nanoparticle (NP) was investigated [21]. Tumor inhibition was observed in the Pan02 orthotopic model, with increased CD8^+^ T cells. However, these studies were performed using mouse allograft models, owing to the intrinsic limitation of the patient-derived xenograft model established using an immunocompromised mouse. To recapitulate human immunity in the patient-derived mouse tumor model, a humanized NSG mouse model was developed and is now commercially available [22,23]. Although the humanized NSG mouse model does not perfectly reproduce the patient’s immune system, it is a valuable in vivo model for testing the agents targeting the tumor immune microenvironment. In the present study, we developed a pancreatic cancer model for the humanized NSG mouse and evaluated the immunotherapeutic effects of siRNA NPs targeting PD-L1, as described later in the paper.

## 2. Materials and Methods

### 2.1. Synthesis of siPD-L1@PLGA NPs

PD-L1 siRNA-loaded poly(lactic-co-glycolic acid) (PLGA) NPs were synthesized via the double-emulsion solvent evaporation (w_1_/o/w_2_) method [19]. PD-L1 siRNAs (50 μg) were complexed with poly-l-lysine (PLL) (100 μg) dissolved in water (200 μL) until the N/P ratio was approximately 1. A gel retardation analysis (1.5% agarose) was performed to confirm a complexing ratio of siPD-L1/PLL (*w*/*w*). The siPD-L1/PLL complexes were mixed with PLGA (20 mg) dissolved in chloroform (2 mL). The mixture was emulsified using a microtip probe sonicator (Branson ultrasonic processor, St Louis, MO, USA) for 1 min. To reduce the surface tension of the PLGA NPs, the primary emulsion solution was mixed with 1% polyvinyl alcohol (PVA) (10 mL) dissolved in distilled water. To generate a double emulsion, the emulsion solution was further emulsified for 2 min. Next, chloroform was evaporated overnight, and then siPD-L1@PLGA NPs collected via centrifugation (16,000× *g*, 1.5 h) were freeze-dried. The siPD-L1 loading efficiency was measured using a Nanodrop spectrophotometer (Thermo Fisher Scientific, Waltham, MA, USA), according to a previously proposed equation [24]. These measurements showed that 2 mg/mL of siRNA@PLGA NPs contained 0.3 mg/mL of siRNA. Additionally, to synthesize polyinosinic-polycytidylic acid sodium salt (poly(I:C))-loaded PLGA NPs, poly(I:C) (100 μg) was complexed with PLL (100 μg) dissolved in distilled water (200 μL). The poly(I:C)/PLL complexes were mixed with PLGA (20 mg) dissolved in chloroform (2 mL). To synthesize tumor lysate-loaded PLGA NPs, the lysed tumor cells (2 mg) were mixed with PLGA (20 mg) dissolved in chloroform (2 mL). The remaining procedures required for the preparation of poly(I:C)@PLGA NPs and tumor lysate@PLGA NPs were similar to those for the siPD-L1@PLGA NP > s.

### 2.2. Derivation of Primary Pancreatic Cancer Cell and Humanized PDX Model

All animal studies were performed under the Guideline for the Care and Use of Laboratory Animals and approved by the Laboratory of Animal Research at the Asan Institute of Life Sciences (project number 2019-14-367). A spontaneous mouse model of pancreatic cancer was generated by crossing a LSL;Kras(G12D) mouse with LSL;Trp53(R172H) [25] and Ptf1a Cre lines. Pancreatic tumors were dissected, and primary cultures were derived as previously described (with clinical information) [26]. For the generation of a humanized PDX model, PDAC tissues successfully grown in an NSG mouse were harvested and minced into 1 mm^3^ tissue fragment. Pieces of the tumor tissue were grafted subcutaneously into humanized NSG mice using a previously described technique [27].

### 2.3. Cell Culture and FACS

Blue #96 and ovalbumin-expressing Blue #96 (Blue-OVA) cells were cultured in Dulbecco’s Modified Eagle’s Medium supplemented with fetal bovine serum (FBS) (10%) and a penicillin-streptomycin solution (1%). The cells were grown in an incubator at 37 °C and 5% CO_2_ until reaching 70% confluency.

### 2.4. Antibodies and Reagents

Chloroform, PVA, PLGA, PLL, and poly(I:C) were obtained from Sigma-Aldrich (St Louis, MO, USA). The following individual primary antibodies were purchased: anti-mouse PD-L1 (Cell Signaling, Danvers, MA, USA) and anti-mouse CD8a (eBioscience, San Diego, CA, USA). PE anti-mouse CD8a, FITC anti-mouse CD8a, FITC anti-mouse PD-L1, and APC anti-mouse INF-γ antibodies were obtained from BioLegend (San Diego, CA, USA). Recombinant murine IL-2 was obtained from PeproTech (Cranbury, NJ, USA). All oligoes were purchased from Bioneer Co. (Tae-Geon, Korea) and the sequences of siRNA were as follows: 5′-GCAGUGACCAUCAAGUCCUdTdT-3′ (human sense siPD-L1), 5′-dTdTAGGACUUGAUGGUCACUGC-3′ (human antisense siPD-L1), 5′-CCUACGCCACCAAUUUCGUdTdT-3′ (scrambled sense siRNA), 5′-dTdTGGAUGCGGUGGUUAAAGCA-3′ (scrambled antisense siRNA).

### 2.5. Cellular Uptake of siPD-L1@PLGA NPs

Blue #96 cells were seeded in 24-well plates, cultured for 24 h, and then transfected with Cy5.5-labeled siPD-L1@PLGA NPs (equivalent to 100 nM Cy5.5-siPD-L1) for 4 h. The cells were washed three times with phosphate-buffered saline (PBS), fixed with paraformaldehyde, stained with 4′,6-diamidino-2-phenylindole (DAPI), and then measured using a laser-scanning confocal microscope (LSM710, Carl Zeiss, Oberkochen, Germany). For a fluorescence-activated cell sorting (FACS) analysis, the washed cells were resuspended in PBS and measured using a Guava EasyCyte flow cytometer (Merck Millipore, Burlington, MA, USA).

### 2.6. Cytotoxicity Study of Scrambled siPD-L1@PLGA NPs on Blue #96 Cells

Varying concentrations of scPD-L1@PLGA NPs (0.06–6.0 mg/mL) or PBS as a control were transfected to Blue #96 cells in 24-well plates (1 × 10^7^ cells/well) for 4 h. After the cells were washed twice with PBS and incubated in a fresh medium for 44 h, a CCK-8 solution (10 μL) was added to each well. After 2 h, the absorbance of the samples was measured at 450 nm using a Spectra MAX 340 Microplate reader (Molecular Device, San Jose, CA, USA).

### 2.7. Isolation of Splenocytes and CD8^+^ T cells

Spleens freshly harvested from naive C57BL/6 mice (6 weeks old, female) were crushed with a plunger and then passed through strainers. To lyse erythrocytes, cell suspensions were reacted with ACK lysis buffer (Thermo Fisher Scientific, Waltham, MA, USA). The lysates were resuspended in an RPMI-1640 medium containing FBS (10%), l-glutamine (2 mM), an ITS liquid media supplement (1%), 2-mercaptoethanol (50 mM), and an antibiotics solution (1%). The CD8^+^ T cells were isolated and purified from the isolated splenocytes by using a CD8a^+^ T-Cell Isolation Kit (Miltenyi Biotec, Bergisch Gladbach, Germany). The isolated CD8^+^ T cells were cultured in 24-well plates (1 × 10^7^ cells/well) in an RPMI-1640 medium containing FBS (10%), l-glutamine (2 mM), an ITS liquid media supplement (1%), 2-mercaptoethanol (50 mM), and an antibiotics solution (1%).

### 2.8. In Vitro Cytolytic Assay of Ovalbumin(OVA)-Specific Cytotoxic T Lymphocytes (CTLs)

To activate OVA-specific CTLs, OT-1 mice were immunized three times at weekly intervals via peritoneal injection of OVA peptide-loaded PLGA (OVApep@PLGA) NPs (200 μg, tumor antigen) and poly(I:C)@PLGA NPs (200 μg, adjuvant). OVA peptide—a class I-restricted epitope of ovalbumin (sequence; SIINFEKL)—was obtained from InvivoGen (San Diego, CA, USA). One week after the last vaccination, spleens were harvested from the immunized mice, and then CD8+ T cells were isolated from the splenocytes via the aforementioned procedures. For re-stimulation, the isolated CTLs were transfected with OVApep@PLGA NPs (1 μg/mL) and mouse IL-2 (50 U/mL) for 1 d. Blue-OVA cells were transfected with siPD-L1@PLGA NPs or PBS for 4 h and incubated for 40 h. The treated Blue-OVA cells (target cells) were stained with CellTracker Deep Red dye (Thermo Fisher Scientific) and subsequently co-cultured with the re-stimulated OVA-specific CTLs (effector cells) in 24-well plates at the indicated effector:tumor (E:T) ratios for 4 h. The fluorescence intensity (FI) derived from the contents of lysed target cells was measured using IVIS Spectrum and IVIS Living Imaging Software (Caliper Life Science Inc., Waltham, MA, USA). The percent-specific lysis was calculated using a previously proposed equation [28].

### 2.9. In Vitro Proliferation Study of OVA-Specific CTLs

By following the aforementioned procedures, OVA-specific CTLs were activated in vivo in OT-1 mice via peritoneal injection of OVApep@PLGA NPs and poly(I:C)@PLGA NPs. Additionally, the isolated OVA-specific CTLs from mice were re-stimulated with OVA peptide and IL-2 in the same procedures. The isolated OVA-specific CTLs were labeled with carboxyfluorescein succinimidyl ester (CFSE). Blue-OVA cells were transfected with siPD-L1@PLGA NPs for 4 h and then incubated for 40 h. Next, the CFSE-OVA-specific CTLs were co-cultured with the treated Blue-OVA cells in 96-well plates at the indicated E:T ratios for 3 d. The proliferation of OVA-specific CTLs was examined using a Guava EasyCyte flow cytometer (Merck Millipore, Burlington, MA, USA).

### 2.10. Production of IFN-γ in Tumor Antigen-Stimulated CTLs

At the end of antitumor experiments involving the humanized, pancreatic PDX model, spleens were collected. CTLs were isolated and re-stimulated with tumor lysate-loaded PLGA NPs using the aforementioned procedures and then cultured in the presence of GolgiPlug^TM^ (BD Biosciences, Franklin Lakes, NJ, USA) for 10 h. After being washed twice with DPBS, the treated CTLs were fixed, permeabilized with a Perm/Wash^TM^ buffer (BD Biosciences), and then stained with FITC-labeled anti-mouse CD8 and APC-labeled anti-mouse IFN-γ antibodies. The production of IFN-γ in the stained CTLs was measured using a Guava EasyCyte flow cytometer.

### 2.11. siPD-L1@PLGA NPs Treatment and Analysis of Tumor-Infiltrated Immune Cells in Humanized NSG Model

The PDAC tumor-bearing humanized mice were injected with vehicle or siPD-L1@PLGA NPs (100 μg/injection) via tail-vein. The nanoparticles were injected twice a week for a total of five times. After 17 days of tumor measurement, the tumor tissues were dissociated using collagenase IV (Thermo Fisher Scientific) and dispase (Thermo Fisher Scientific). After lysing red blood cells (RBCs), the cells were counted. The single-cell suspension was stained for human CD45 (BioLegend, San Diego, CA, USA, cat no. 304018), hCD3 (BioLegend, cat no. 300320), and hCD19 (BioLegend, cat no. 560994), followed by flow cytometry (Accuri C6, BD, Franklin Lakes, NJ, USA). To assess the human lymphocyte composition in the blood of humanized mice, the blood was collected, and RBCs were lysed. The single-cell suspension was stained for human CD3 (BioLegend, cat no. 344805), hCD19 (BD, cat no. 560994), and CD45 (BioLegend, cat no. 304018), followed by flow cytometry.

### 2.12. Measurement of Lymphocyte-Mediated Cytotoxicity from Tumor-Bearing Mouse

For the experiment involving the lymphocyte-mediated cytotoxicity to tumors, splenocytes were isolated from the humanized mice bearing PDAC cells. After lysing RBCs, the single-cell suspension was placed in the plate coated with human CD3 and CD28 antibodies. PDAC cells were prepared after 72 h of treatment with vehicle or siPD-L1@PLGA NPs (2 µg/mL). The activated splenocytes were co-cultured with siPD-L1@PLGA NP-treated PDAC cells at an E:T ratio of 20:1 and incubated at 37 °C for 24 h. The cells were collected and stained for human E-cadherin (Cell Signaling Technology, Danvers, MA, USA, 24E10), together with Annexin V (BD, cat no. 80-1729) and PI (ENZO, Farmingdale, NY, USA, cat no. 80-1731), followed by flow cytometry.

### 2.13. Multiplexed Fluorescent Immunohistochemistry

Four-micron-thick slices were cut from the tissue and transferred onto positively charged slides, followed by multiplexed fluorescent immunohistochemistry with a Leica Bond Rx™ Automated Stainer (Leica Biosystems, Nussloch GmbH, Nussloch, Germany). The slides were baked at 60 °C for 40 min and deparaffinized with a Leica Bond Dewax solution (Cat #AR9222, Leica Biosystems, Nussloch, Germany), followed by antigen retrieval with Bond Epitope Retrieval 2 (Cat #AR9640, Leica Biosystems) for 30 min. After the antigen retrieval, the slides were incubated with primary antibodies followed by a secondary horseradish peroxidase-conjugated polymer. Each horseradish peroxidase-conjugated polymer led to the covalent bonding of a different fluorophore using tyramide signal amplification. This covalent bonding was followed by additional antigen retrieval with Bond Epitope Retrieval 1 (Cat #AR9961, Leica Biosystems, Milton Keynes, UK) for 20 min to remove prior primary and secondary antibodies before the next step in the sequence. Each slide was subjected to six sequential rounds of staining. After the sequential reactions, sections were counterstained with Spectral DAPI and mounted with HIGHDEF® IHC fluoromount (Enzo Life Sciences, Farmingdale, NY, USA).

The sections were stained using an Opal Polaris 7-Color Automated IHC Detection Kit (AKOYA Biosciences, Marlborough, MA, USA). Cells were stained with antibodies against CD4 (1:100, Abcam, Cambridge, UK), CD8 (1:300, AbD Serotec, Hercules,CA, USA), Foxp3 (1:100, Abcam), PD-L1 (1:300, CST, Danvers, MA, USA), GranzymeB (1:50, CellMarque, Rocklin, CA, USA), and CD45RO (1:13500, CST), and the fluorescence signals were captured with the following fluorophores: Opal 520, Opal 540, Opal 570, Opal 620, Opal 690, and Opal 780.

### 2.14. Multispectral Imaging and Analysis

Multiplex stained slides were scanned using a Vectra^®^ Polaris Quantitative Pathology Imaging System (Akoya Biosciences, Marlborough, MA/Menlo Park, CA, USA), and images were visualized in the Phenochart whole slide viewer (Akoya Biosciences, Marlborough, MA/Menlo Park, CA, USA). The images were analyzed using the inForm 2.4.4 image analysis software (Akoya Biosciences, Marlborough, MA, USA/Menlo Park, CA, USA) and Spotfire (TIBCO Software Inc., Palo Alto, CA, USA).

### 2.15. DLS Analysis of siRNA@PLGA NPs

The dynamic diameter of zeta potential of empty PLGA NPs and siRNA@PLGA NPs were measured using a Malvern Nano ZS and Zeta-sizer (Malvern Instrument, Malvern, UK). Samples were serially diluted and each data were collected at a scattering angle of 173° with a 633 nm laser.

### 2.16. Statistics

All data are presented as the mean ± standard deviation (SD). Analysis between groups was performed using the Student’s *t*-test. The *p*-values of <0.05, <0.01, and <0.001 were denoted as *, **, and ***, respectively.

## 3. Results

### 3.1. Synthesis of siRNA Nanoparticles Targeting PD-L1 and In Vitro Validation

For the functional evaluation of PD-L1-targeting siRNA NPs in pancreatic cancer, we first isolated primary cancer cells from a spontaneous mouse model of pancreatic cancer [25] (called Blue cell, Figure 1A, left and middle panels). The PDAC model has active Kras mutation (G12D) and dominant-negative Trp53 mutation (R172H) that are conditionally expressed by Cre under the control of pancreatic specific promoter Ptf1a [29]. The genotypes of three mutations were confirmed (Figure 1A, right panels).

Based on the dynamic light scattering analysis, the particle sizes of empty PLGA NPs and siRNA@PLGA NPs were 174.8 ± 2.4 and 188.5 ± 1.2 nm, respectively (Figure 1B). The negative charge in the empty PLGA NPs (−5.552 mV) became slightly neutralized in siRNA@PLGA NPs (−3.364 mV) after the positively charged PLL/siRNAs were complexed. Next, siRNA for PD-L1 encapsulated in NPs (siPD-L1@PLGA) efficiently suppressed the PD-L1 expression of the cell, at both the RNA (Figure 1C) and protein levels (Figure 1D), when compared to only PBS-treated control after IFN-γ stimulation. As expected, the scrambled siRNA nanoparticles (scPD-L1@PLGA) showed no suppression of PD-L1 expression at both RNA and protein levels, similar to the untreated control (data not shown). Up to 6 mg/mL, no toxic effect of the scrambled scPD-L1@PLGA was observed (Figure 1E). When the concentration of scPD-L1@PLGA increased to 12 mg/mL, cell viability was about 84% (data not shown). Given that the non-cytotoxic concentration range is defined as greater than 90% of cell viability, these results indicate that the concentration ranges below 6 mg/mL do not induce any cytotoxic effect in Blue #96 cells. We selected 2 mg/mL as an optimized concentration for in vitro experiments.

Microscopic imaging of florescent dye-labeled NPs indicated robust uptake by the cells at a concentration of 2 mg/mL (Figure 2A). An FACS analysis also indicated efficient cellular uptake of the NPs (Figure 2B). Next, we monitored the time-dependent change in the PD-L1 protein level after siPD-L1@PLGA treatment. The western blot data shown in Figure 2C indicate a significant reduction in the PD-L1 level after 2–3 d of treatment. Furthermore, the FACS analysis revealed that the siPD-L1@PLGA downregulated the IFN-γ-induced PD-L1 expression, as shown in Figure 2D. As expected, the scrambled scPD-L1@PLGA showed no downregulation of IFN-γ-induced PD-L1 expression. These data collectively indicate the efficient knockdown of the PD-L1 expression in pancreatic cancer cells by siPD-L1@PLGA.

### 3.2. siPD-L1@PLGA Abrogates Immune Escape Function of Pancreatic Tumor Cells

To determine whether siPD-L1@PLGA NPs reactivate the cytotoxicity of CTLs, we generated a pancreatic cancer cell line with the stable expression of ovalbumin (Blue-Ova, Figure 3A). Additionally, we re-stimulated OVA-specific CD8^+^ T cells in the manner described in Methods and transfected Blue-OVA cells in parallel with siPD-L1@PLGA NPs. For immune challenge, we co-cultured the stimulated CD8^+^ T cells with the transfected Blue-OVA cells stained using CellTracker Deep Red dye (E:T ratios of 1:1 and 5:1). According to the FI of the lysed cell contents, the siPD-L1@PLGA-treated sets (ii–iv) exhibited increased cytotoxicity of CTLs against Blue-OVA cells at both the 1:1 and 5:1 ratio, compared with the only PBS-treated control set without immunization (Figure 3B,C). As expected, the scrambled siPD-L1@PLGA-treated sets did not show an increase in the cytotoxicity of CTLs against Blue-OVA cells at both ratios, similar to the PBS-treated sets (data not shown). These results imply that inhibition of PD-1/PD-L1 interactions via RNAi enhances the cytotoxicity of CTLs.

To investigate whether silencing of PD-L1 on cancer cells promotes proliferation of tumor-specific CTLs, we re-stimulated OVA-specific CD8^+^ T cells and transfected Blue-OVA cells with siPD-L1@PLGA NPs in the manner described above. Next, we co-cultured CFSE-labeled CD8^+^ T cells with Blue-OVA cells at different E:T ratios. An FACS analysis indicated that the silencing of PD-L1 on the Blue-OVA cells significantly increased the proliferation of CTLs at three different E:T ratios, in contrast to those of an untreated control set (Figure 3D,E). As expected, the scrambled siPD-L1@PLGA-treated Blue-OVA cells did not show increased proliferation of CTLs at three different E:T ratios, similar to the untreated control set (data not shown). These results indicate that inhibition of PD-1/PD-L1 interactions via RNAi leads to local expansion of tumor-specific CTLs.

### 3.3. Efficacy Test in Humanized PDX Model Reveals Antitumor Effect of siPD-L1@PLGA via Immunomodulation

To evaluate the antitumor effect of the siPD-L1@PLGA in a preclinical model, we introduced humanized NSG mice that contained approximately 30% of human CD45^+^ cells in PBMC (Appendix A for human immune cell contents). These mice were implanted subcutaneously with PDAC patient-derived xenograft tumor established previously [26]. We selected primary PDAC (labeled as 19224) with high PD-L1 (Appendix A) and availability. The tumor growth was monitored with the injection of siPD-L1@PLGA twice a week. The injection of siPD-L1@PLGA NPs caused no significant reduction in body weight, indicating that there was no severe cytotoxicity (Appendix A). Periodic monitoring of the tumor volume indicated that the siPD-L1@PLGA treatment significantly suppressed the PDAC growth until the end of the experiments (Figure 4A and Appendix A, H&E staining of each tumor is shown in Appendix A). Next, the dissected tumors were subjected to a FACS analysis for profiling the infiltrated immune cells (Appendix A). The siPD-L1@PLGA-treated mice exhibited more tumor-infiltrated lymphocytes (TILs) than the only vehicle-treated control mice (although the difference was not statistically significant; see Section 4), as evidenced by an increased CD45^+^CD3^+^ (T cells) or CD45^+^CD19^+^ (B cells) population (Figure 4B for count and Appendix A for composition, which was increased from 5.6% to 8.0%). Consequently, the blood lymphocyte count was reduced (Appendix A). Importantly, we observed significantly more IFN-g positive, activated CD8 cells after the treatment of siPD-L1@PLGA (Figure 4C). An Annexin V/PI analysis of E-cadherin positive (PDAC marker) cells co-cultured with splenocytes from each mouse group indicated that the apoptotic population of tumors was increased by the siPD-L1@PLGA treatment, validating the antitumor effect (17.2% in control, 33.3% in siPD-L1@PLGA for Annexin V-positive cells; Figure 4D). These results confirm that the siPD-L1@PLGA abrogates pancreatic tumor growth by increasing and activating TIL through the inhibition of PD-1/PD-L1 interactions, which induces apoptosis of cancer cells.

### 3.4. OPAL Multiplex IHC Analysis Reveals Granzyme B Upregulation in siPD-L1@PLGA-Treated Tumors

To further investigate the modulation of the tumor immunity by the siPD-L1@PLGA NPs, we analyzed the tumors using a multiplex immunohistochemistry system (OPAL) [30]. We selected markers, including CD45RO (activated and memory T lymphocytes), CD4 and CD8 (T cell), FoxP3 (Treg cell), Granzyme (activated NK and T cell), and PD-L1, to simultaneously monitor the changes in multiple immune cells due to siPD-L1 NPs. Figure 5A shows whole-tumor images and six grids indicating randomized selection of the fields to be analyzed. Figure 5B,C shows representative fluorescence (top images) and colored images used for the quantitation of each selected marker (lower images; see Appendix A for raw data). The quantitation of the signal for selected markers revealed comparable numbers of immune cells in the control and siPD-L1@PLGA NP-treated tumors (Figure 5D, Appendix A). Overall, the expression of Granzyme B was higher for the siPD-L1 treated tumors, although the difference was not statistically significant, owing to the high variability (Figure 5E; see Discussion).

## 4. Discussion

Our results indicate the clinical potential of siRNA-mediated PD-L1 knockdown, which suppresses pancreatic tumor growth in the humanized preclinical model. The humanized NSG mouse is generated by implanting Cd34^+^ hematopoietic stem cells in a young female NSG mouse. Owing to the human leukocyte antigen (HLA) mismatching, the tumor graft can be rejected in the humanized NSG model [31]. Even though there was no HLA genotyping information available for the commercial humanized NSG mouse, we could establish one of the PDAC tumors in this model. However, we observed variability of the T cell as well as other marker profiles in both the control and siPD-L1@PLGA-treated groups (Figure 5E and Appendix A), making the statistical analysis difficult. We cannot exclude the possibility that the high variability of the T-cell immunity in our humanized PDAC mouse model is affected by the HLA mismatch. To circumvent this problem, several groups adopted the patient-derived PBMCs as an immune cell source [32]. However, the variability of the success rate, which is largely dependent on the donor, and the short lifespan of the PBMCs in mice limit this approach. Further studies are required to resolve the HLA mismatch between CD34+-driven host immune cells and the grafting tumor. Additionally, the consistency of immune cell function in the humanized NSG mouse must be confirmed.

The siPD-L1@PLGA enters cancer cells and inhibits the PD-L1 expression efficiently in vitro (Figure 1 and Figure 2). Studies on the therapeutic potential of PD-L1 suppression via RNAi have been published recently in hepatocellular carcinoma [33] and triple-negative breast cancer [34]. Considering the tissue-delivery advantage of the siPD-L1@PLGA compared with naked siRNA, siPD-L1@PLGA is expected to be applied to the stomal-rich PDAC model. Indeed, we generated an orthotopic PDAC model by injecting patient-derived cells with the stable expression of luciferase, which allowed us to detect the growing tumor using bioluminescence. Although bioluminescence imaging was sensitive enough to detect tumors growing in the pancreas, the ROI (Region of Interest) value (reflecting the bioluminescence intensity) fluctuated, probably owing to the inconsistent depth of the pancreas and mobility of the tissue (in the mouse abdomen) during imaging, which significantly affected the signal (data not shown). Hence, we presented the efficacy of the siPD-L1@PLGA in a subcutaneous model. In the future, the variability of tumor growth in an orthotropic model can be minimized by adopting a more precise surgical technique as well as increasing the number of mice in each group.

Despite the limitations of the present study, the siPD-L1@PLGA is promising for PDAC immunotherapy, as it exhibited low toxicity (Appendix A) and is easy to generate with a relatively low cost. Further study involving combination with standard chemotherapy or the establishment of criteria for screening applicable patient groups will facilitate the clinical application of this agent in the near future.

## Figures and Tables

**Figure 1 cells-10-02734-f001:**
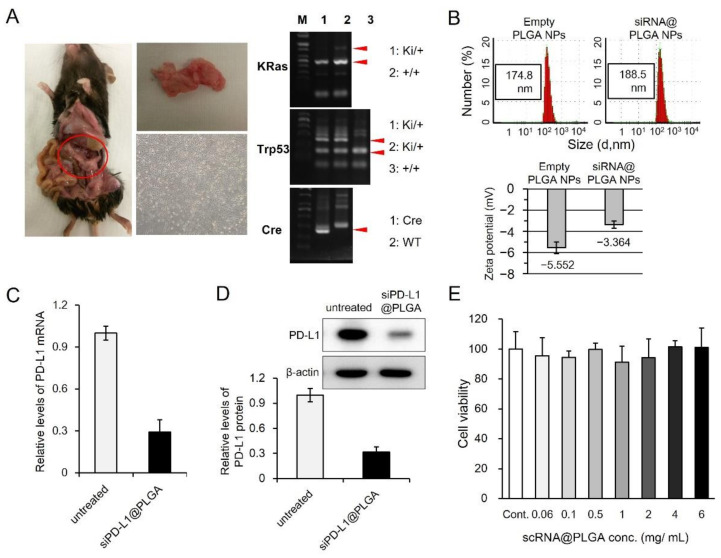
siPD-L1@PLGA suppresses PD-L1 expression in pancreatic cancer cells without toxicity. (**A**) (left panels) Representative photographs of a pancreatic tumor and primary cells isolated from the KRasG12D; Trp53R172H; Ptf1aCre mouse model. (Right panels) Genotyping results confirming KRasG12D (top), Trp53R172H (middle), and Ptf1aCre (bottom). (**B**) DLS analysis of empty PLGA NPs and siRNA@PLGA NPs. Particle size and zeta potential were presented as the mean ± SD (*n* = 3). (**C**,**D**) In vitro silencing of PD-L1 in the siPD-L1@PLGA-treated Blue #96 cells. Cells stimulated with IFN-γ for 4 h were transfected with siPD-L1@PLGA NPs for 4 h and then cultured for 68 h. The mRNA and protein levels of PD-L1 were measured via qRT-PCR (**C**) and western blotting (**D**), respectively. The untreated samples exhibited IFN-γ-stimulated cells without siPD-L1@PLGA transfection. The results are presented as the mean ± SD (*n* = 3). (**E**) Cell viability of scrambled siPD-L1@PLGA-treated Blue #96 cells. The cytotoxicity of scPD-L1@PLGA NPs was analyzed via a CCK-8 cytotoxicity assay. The results are presented as the mean ± SD (*n* = 3).

**Figure 2 cells-10-02734-f002:**
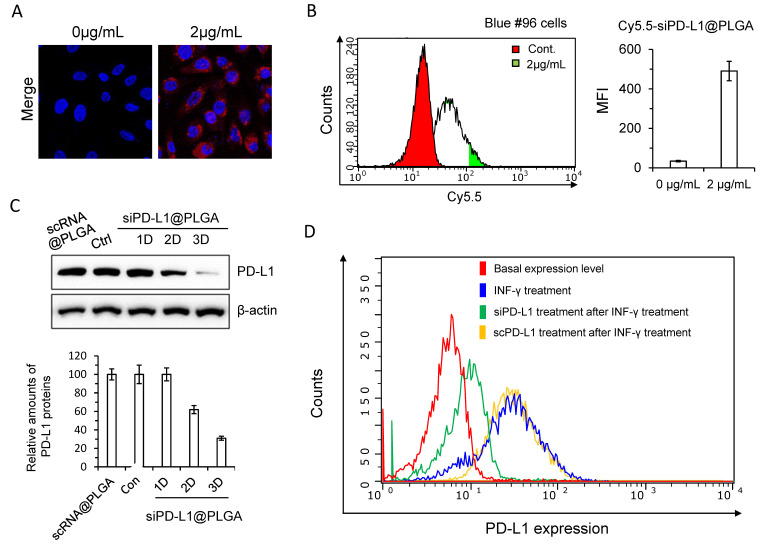
siPD-L1@PLGA efficiently enters and suppresses IFN-induced PD-L1 of PDAC cells. (**A**) Cellular uptake of Cy5.5-scRNA@PLGA NPs in the Blue #96 cells examined using confocal microscopic images. Cells were transfected with Cy5.5-scRNA@PLGA NPs for 4 h, and then their fluorescence images were measured. The nuclei were stained with DAPI dyes (blue). Red signals indicate Cy5.5-scRNA. The results are presented as the mean ± SD (*n* = 3). (**B**) FACS histogram of Cy5.5-scRNA@PLGA-treated Blue #96 cells. Cells were transfected with Cy5.5-scRNA@PLGA NPs for 4 h and then analyzed against a prefixed gate region for Cy5.5 dyes. The results are presented as the mean ± SD (*n* = 3). (**C**) Western blot images of Blue #96 cells after siPD-L1@PLGA NPs transfection. IFN-γ-stimulated Blue #96 cells were transfected with siPD-L1@PLGA NPs for 4 h and incubated for the indicated period. The PD-L1 protein levels were analyzed using the western blotting method. The control cells were IFN-γ-stimulated cells without transfection. The PD-L1 protein levels of the control cells and scRNA@PLGA-treated cells were measured 3 days after transfection. The relative protein levels of PD-L1 are plotted at the bottom. The results are presented as the mean ± SD (*n* = 3). (**D**) FACS analysis indicated suppression of the PD-L1 expression on siPD-L1@PLGA-treated Blue #96 cells under IFN-γ stimulation. Cells were stimulated and transfected in a manner similar to that for Figure 1B. As a control, PD-L1 expression on scPD-L1@PLGA-treated Blue #96 cells under IFN-γ stimulation was shown.

**Figure 3 cells-10-02734-f003:**
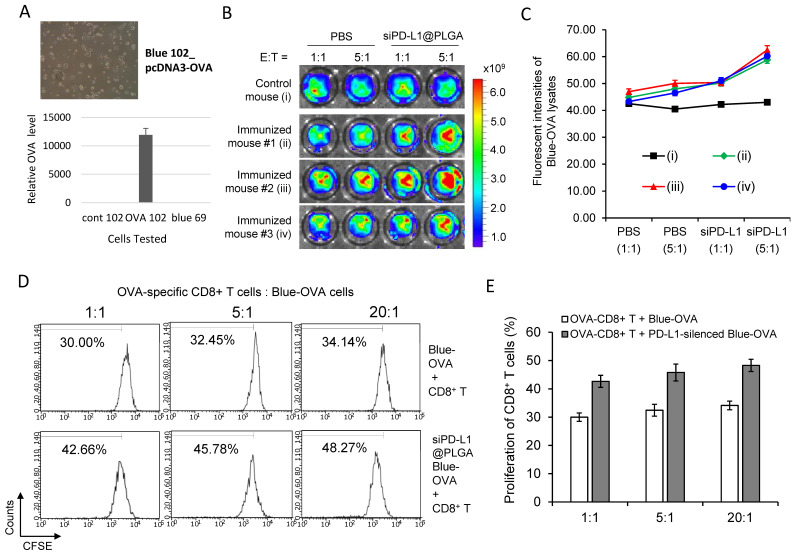
In vitro efficacy of siPD-L1@PLGA in OVA-specific T-cell immunity. (**A**) Generation of PDAC cells with stable OVA expression. The left panel shows a photograph of the OVA cells, and the right panel shows the Ova RNA level. (**B**,**C**) Cytolytic activity of OVA-specific CD8^+^ T cells exhibiting antitumor T-cell immunity. OVA-specific CD8^+^ T cells were isolated from the immunized OT-1 mice with the OVA peptide (SIINFEKL) and adjuvant (ii–iv) and then re-stimulated using OVA peptide-loaded PLGA NPs and recombinant mouse IL-2. Blue-OVA cells were transfected with siPD-L1@PLGA NPs or PBS for 4 h and incubated for 40 h. Next, the treated Blue-OVA cells (target cells) were stained with CellTracker Deep Red dye and then co-cultured with OVA-specific CD8^+^ T cells (effector cells) at a ratio of 1:1 or 1:5 (E:T) for 4 h. The FI derived from the contents of lysed Blue-OVA cells was measured (**B**), and the percent specific lysis was calculated and plotted (**C**). OVA-nonspecific CD8^+^ T cells isolated from non-immunized C57BL/6 mice (i) were used as a control. The results are presented as the mean ± SD (*n* = 3). (**D**,**E**) Promoted expansion of CTLs induced by silencing of PD-L1 in the co-culture system of OVA-specific CD8^+^ T cells and Blue-OVA cells. OVA-specific CD8^+^ T cells were isolated from the in vivo-immunized OT-1 mice with the OVA peptide and adjuvant and then re-stimulated. Blue-OVA cells transfected with siPD-L1@PLGA NPs were co-cultured with CFSE-labeled OVA-specific CD8^+^ T cells at an E:T ratio of 1:1 or 1:5 for 3 d. The proliferation of CTLs was examined via FACS analysis (**D**) and then plotted in comparison with the co-culture samples without siPD-L1@PLGA transfection (**E**). The results are presented as the mean ± SD (*n* = 3).

**Figure 4 cells-10-02734-f004:**
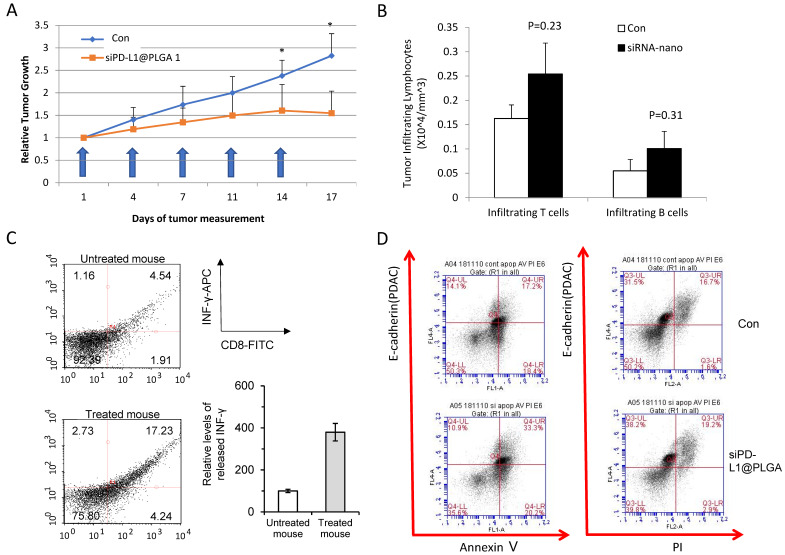
siPD-L1@PLGA suppressed PDAC growth in the humanized NSG mouse model. (**A**) Graph showing the growth of control (PBS, in blue) and siPD-L1@PLGA-treated (orange) PDAC in the humanized NSG mouse. Blue arrows indicate siPD-L1@PLGA injection. The *p*-values of <0.05 was denoted as *. (**B**) Tumor-infiltrating lymphocytes. Densities of T cells (hCD45^+^hCD3^+^) and B cells (hCD45^+^hCD19^+^) in the PDAC tumor burden. Data are expressed as the mean ± SD (*n* = 4–5 mice/group). “ns” indicates a “not significant” result for the two-tailed unpaired Student’s *t*-test. (**C**) FACS histograms for the production of IFN-γ in the tumor antigen-stimulated CD8^+^ T cells. The isolated CD8^+^ T cells from siPD-L1@PLGA-treated mice were re-stimulated with tumor-loaded PLGA NPs and then stained with FITC-labeled anti-mouse CD8 and APC-labeled anti-mouse IFN-γ antibodies, followed by FACS analysis. The relative levels of released IFN-γ were plotted in comparison with those for untreated mice. The results are presented as the mean ± SD. (*n* = 6). (**D**) Representative flow cytometry plots of the cytotoxicity (PI/Annexin V double positivity in E-cadherin^+^ PDAC cells) mediated by splenocytes obtained from tumor-bearing mice. The histograms on the left and right correspond to the control and siPD-L1@PLGA-treated mice, respectively. Note the increased portion of the first quadrant, indicating PI/Annexin V-positive cells.

**Figure 5 cells-10-02734-f005:**
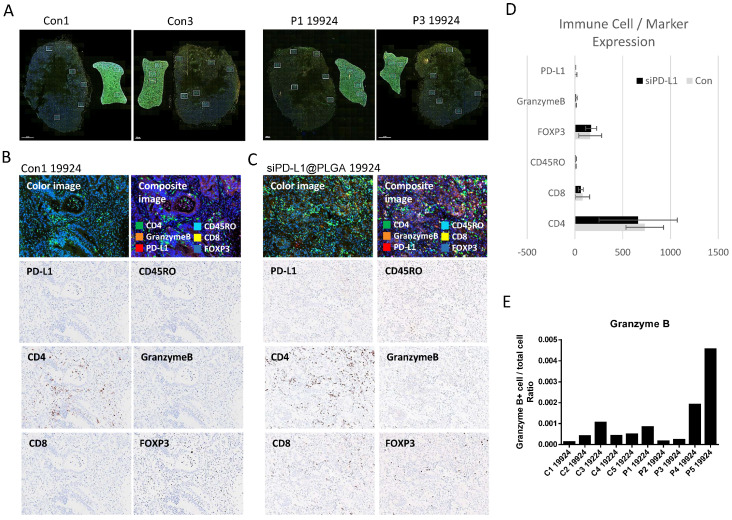
Multiplex immunohistochemistry analysis (OPAL) revealed that the tumor-immune microenvironment was modulated by siPD-L1@PLGA. (**A**) Representative whole tumor images for the control tumors (C1 and C3) or siPD-L1@PLGA-treated tumors (P1 and P3) from the humanized NSG mouse. (**B**,**C**) Specific IHC images of the control tumors (**B**) and siPD-L1@PLGA-treated tumors (**C**). PD-L1, CD45RO (Activated and memory T lymphocytes), CD4 and CD8 (T cell), FoxP3 (Treg cell), and Granzyme B (Activated NK and T cell) were tested. (**D**) Quantitation results for the markers shown in B and C (*n* = 5 for each group). (**E**) Individual signal intensity of the Granzyme B in control or siPD-L1@PLGA-treated tumors.

## Data Availability

The raw data of this article is available in online Appendix A.

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
