# Peer review of "siRNA Nanoparticle Targeting PD-L1 Activates Tumor Immunity and Abrogates Pancreatic Cancer Growth in Humanized Preclinical Model"

_cells, 2021, doi:10.3390/cells10102734_

Round 1

Reviewer 1 Report

In their manuscript, Jung et al., describe the potential of a siRNA nanoparticle targeting PD-L1 for increasing tumor immunity and abrogating pancreatic cancer growth. The authors demonstrate that knockdown of PD-L1 per sé does not have an effect on PDAC cells, but successfully sensitizes them to antigen-specific immune cells. Using a humanized NSG model transplanted with a PDAC PDX model the authors´ findings further suggest that PDAC growth can be abolished upon siRNA PD-L1 application which might be associated with a higher activity of B- and NK cells in the mice.

The data depicted here are interesting and solid and experiments are controlled and interpreted in an appropriate way. For the humanized model, just one PDAC patient tumor is utilized which, based on the heterogeneity of the disease, limits conclusions on the efficacy of the targeting strategy in PDAC in general. Nevertheless, this is a nice and important proof-of-principle study not only suggesting nanoparticle-based PD-L1 targeting for further exploration in PDAC therapy, but also indicating the advantages and disadvantages of humanized models for studying this disease.

Specific points:

  • Which recipient cells have been used to generate Blue 102_pcDNA3-OVA cells?
  • What is the measurement unit in Fig. 4A?
  • For the humanized models: please indicate the way of transplantation. The title states orthotopic, but this is not mentioned in the results. The scale and tumor diameter rather suggests subcutaneous transplantation?
  • It would be nice to show H&E stainings of the vehicle- or siPD-L1 treated mice in order to confirm the PDAC histology.

Author Response

Point by Point responses to reviewer’s comments. (cells-1381872)

Reviewer #1

In their manuscript, Jung et al., describe the potential of a siRNA nanoparticle targeting PD-L1 for increasing tumor immunity and abrogating pancreatic cancer growth. The authors demonstrate that knockdown of PD-L1 per sé does not have an effect on PDAC cells, but successfully sensitizes them to antigen-specific immune cells. Using a humanized NSG model transplanted with a PDAC PDX model the authors´ findings further suggest that PDAC growth can be abolished upon siRNA PD-L1 application which might be associated with a higher activity of B- and NK cells in the mice.

The data depicted here are interesting and solid and experiments are controlled and interpreted in an appropriate way. For the humanized model, just one PDAC patient tumor is utilized which, based on the heterogeneity of the disease, limits conclusions on the efficacy of the targeting strategy in PDAC in general. Nevertheless, this is a nice and important proof-of-principle study not only suggesting nanoparticle-based PD-L1 targeting for further exploration in PDAC therapy, but also indicating the advantages and disadvantages of humanized models for studying this disease.

>> We sincerely appreciate the reviewer’s positive response to our manuscript and respectfully addressed the valuable comments as follows.

Specific points:

  • Which recipient cells have been used to generate Blue 102_pcDNA3-OVA cells?

>> We used Blue 102 cells as a recipient cell. The Blue cells are derived from a spontaneous model of PDAC, generated in our previous study (unpublished).  As shown in Figure 1A, it has KrasG12D and Trp53 R172H mutation specifically expressed in pancreatic tissue by Ptf1a-Cre. We added this comments in methos section 2.2 and 2.3

  • What is the measurement unit in Fig. 4A?
  • >> We thank for the critical comment. The data in Figure 4A shows a relative tumor growth, measured by calipers. The unit is mm^3 (volume). In fact, we also tried to measure the PDAC tumor growth from IVIS, using a luciferase labelled, orthotropic mouse model. However, due to the high variability of basal ROI signal at the point of siRNA nanoparticle injection, we switched to subcutaneous model and obtained tumor growth data shown in Figure 4. We corrected this (along with the title) and clarified the data is generated from a subcutaneous model.

  • For the humanized models: please indicate the way of transplantation. The title states orthotopic, but this is not mentioned in the results. The scale and tumor diameter rather suggests subcutaneous transplantation?

>> AS the reviewer assumed, the data we presented here is obtained from subcutaneous model. We regret our mistake in the title and manuscript (discussed in the answer above). We deeply thank for the reviewers’ comment to correct our mistake before publication.

  • It would be nice to show H&E stainings of the vehicle- or siPD-L1 treated mice in order to confirm the PDAC histology.

>> Following the comment, we added H&E staining data of vehicle- or siPD-L1 treated tumors in the supplementary Figure 2C. As the tumors were obtained from subcutaneous model, the histology indicated high portion of tumor cells.

Reviewer 2 Report

In here the authors introduced a poly(lactic-co-glycolic acid;PLGA)-based siRNA nanoparticle targeting PD-L1 (siPD-L1@PLGA) that overcomes the problem of drug delivery into the fibrotic PDAC tumor.

The following are my comments and critique:

Minor:

  1. Introduction.  The authors should provide background on the PLGA nanoparticles used and how is this been utilized as carrier for drug delivery in PDAC or other forms of cancers.
  2. Methods 2.2. It is unclear whether the PDAC cells used to inoculate to humanized NSG mice were from GEMM or from PDAC patient tissue/cells. As this is a humanized mouse, I assumed that the PDAC should be sourced from human patient tissues (as the name imply PDX). Please revise the wording from your Methods [Pancreatic tumors were dissected, and primary cultures were derived as previously described [21]. For the generation of a humanized PDX model, PDAC tissues successfully grown in an NSG mouse were harvested and minced into 1-mm3 tissue fragment. Pieces of the tumor tissue were grafted onto the pancreatic tails of humanized NSG mice via a tissue adhesive technique [22]]. Also, in this section, please detail/elaborate that you have successfully produced PDX from previous studies and you have utilized those primary cells in your humanized PDX model.
  3. Methods 2.3. Describe how the CD8+ T cells were isolated and from which source (I am aware that you have described how you have isolated the CD8+ T cells but since this method comes before that (Methods 2.7., I think it is appropriate that you mention the details in this section as well).
  1. The authors have mentioned in Figure 4C that they have significantly showed increase level of CD8 and IFN-y but it’s not reflected in their graph. Also, line 342-346 or in the discussion, the author should mention the significant of this specific experiment or why did they measure IFN-y secretion and correlate this to the function of PD-L1 in the tumor (inhibit the activity and toxic effects of CD8+T cells). Also, in Figure 4B:  the author should add what is the percentage of CD8+ T cells from the infiltrating T lymphocyte?
  2. Figure 5: Why did the authors opt to randomly select 6 grids for analysis? What happen if the authors select the whole area (tumor) and analyze based on cell numbers (in this case, immune cell infiltrates? Or could the authors focus on the area where there are more immune infiltrates and analyse from there the markers (CD45RO (Activated and memory T lymphocytes), CD4 and CD8 (T cell), FoxP3 (Treg cell), Granzyme (Activated NK and T cell), and for PD-L1 (whole tumor), as I understand there is tumor heterogeneity that’s why the results are not significant.
  3. Discussion. Mention how your results compare to other studies on RNAi-Mediated PD-L1 in Pancreatic Cancer Immunotherapy. 
  4. The authors should be consistent in labelling experimental groups (i.e. Figure 4 Ct/ Cont/ Control).
  5. The authors should edit the following lines (for grammar):

Line 113-114

Line 263

Line 339: vesicle-treated mice or vehicle-treated mice

Major:

1.   Methods/Results. The authors failed to detail the characteristics of siRNA-loaded PLGA NPs in terms of size, zeta potential. Also, they did not mention how did they account or measure encapsulation efficiency and how do they know if they are administering same amount to each mouse (siRNA loaded PLGA NPs)? Also, it is not clear whether the control group (in vivo) is PBS only or a scrambled siRNA -loaded PLGA. The authors should provide the sequences of siRNAs used.

2. Figure 1: It is unclear whether the authors treat the primary cells with scPD-L1 (served as control/untreated) for Figure 1B-1C or is it only stimulation with IFN then leave it for 68h without scrambled siRNA? If the authors didn’t, they should include an appropriate control such as the scrambled siRNA. For Figure 1D. Where is the cytotoxicity for siPD-L1?

3. For Figure 2D, where is the scRNA-PLGA group like in Figure 2C?

4. Figure 2B-2E: The authors should include group with scrambled siRNA and not PBS only as their appropriate control to have a direct comparison and rule out the effect of PD-L1 (knockdown) for enhanced cytotoxicity of CTLs.

5. It is unclear whether the Ct group in Figure 4 is PBS treated mice or includes scrambled siRNA/PLGA as control. Also, mention how did the authors administer (route of administration) the siPD-L1@PLGA to the animals? And how much of siPD-L1 and PLGA NPs? When did they start their treatment?

Finally, the authors should add more animal numbers for their humanized PDX model; check the expression of PD-L1 in their patient derived cell cultures (PDX that they have used to inoculate to humanized NSG mice). Discussion: Perhaps, they need to include in the discussion the history of the patient/s where they derived the PDX lines (any prior drug treatment, resistance to any treatments, PD-L1 status) to correlate with their findings. Since the humanized NSG model is not the best model to use for this study, perhaps they can do proof of concept experiments by growing these PDX lines (PDAC) in different matrices or in 3D co-cultured with CAFs, stellate cells or immune cells (PBMC) and test their siRNA-PLGA in combination with another therapy/drug.

Author Response

Point by Point responses to reviewer’s comments. (cells-1381872)

Reviewer #2

In here the authors introduced a poly(lactic-co-glycolic acid;PLGA)-based siRNA nanoparticle targeting PD-L1 (siPD-L1@PLGA) that overcomes the problem of drug delivery into the fibrotic PDAC tumor.

>> We sincerely appreciate the reviewer’s comments to our manuscript and respectfully addressed the valuable comments as follows.

The following are my comments and critique:

Minor:

  1. The authors should provide background on the PLGA nanoparticles used and how is this been utilized as carrier for drug delivery in PDAC or other forms of cancers.

>> We agree that our introduction for PLGA nanoparticle was insufficient. Hence, we added this paragraph in the introduction part.

PLGA polymers have widely provided efficient drug delivery carriers for chemotherapeutics and nucleotides, due to their low cytotoxicity, biodegradability, sustained release property, and enhanced permeability and retention (EPR) effect in the medical applications for cancer treatment [11, 12, 13, 14]. Indeed, the Food and Drug Administration have been approved several PLGA formulations for drug delivery in humans [104]. Thus, PLGA nanoparticles as siRNA delivery vehicles have drawn great potential in the RNAi-mediated therapeutic applications, in contrast to the commonly used polycationic carriers, which inevitably cause cytotoxic and/or non-degradable issues [11].”.

  1. Methods 2.2. It is unclear whether the PDAC cells used to inoculate to humanized NSG mice were from GEMM or from PDAC patient tissue/cells. As this is a humanized mouse, I assumed that the PDAC should be sourced from human patient tissues (as the name imply PDX). Please revise the wording from your Methods [Pancreatic tumors were dissected, and primary cultures were derived as previously described [21]. For the generation of a humanized PDX model, PDAC tissues successfully grown in an NSG mouse were harvested and minced into 1-mm3 tissue fragment. Pieces of the tumor tissue were grafted onto the pancreatic tails of humanized NSG mice via a tissue adhesive technique [22]]. Also, in this section, please detail/elaborate that you have successfully produced PDX from previous studies and you have utilized those primary cells in your humanized PDX model.

>> We thank for the critical comment and agree that the description was confusing. In fact, the humanized PDX was generated from PDAC patients tissue. We tried both orthotropic and subcutaneous model, but the tumor growth in orthotropic model was inconsistent. The data in Figure 4 represent subcutaneous model. We regret the methods section was not properly revised, and corrected it as follows.

>> Pieces of the tumor tissue were grafted subcutaneously into humanized NSG mice using a previously described technique [27].

Also, we revised results section 3.3 as follows

>> These mice were implanted with PDAC patient-derived xenograft tumor established previously [26]. The PDAC cell from GEMM (Named as Blue cell) was used for the data presented in the Figure 1-3. As this is a PDAC cell derived from immune competent mouse, we were able to perform in vitro cytotoxicity assay using Ova-T cells

  1. Methods 2.3. Describe how the CD8+ T cells were isolated and from which source (I am aware that you have described how you have isolated the CD8+ T cells but since this method comes before that (Methods 2.7., I think it is appropriate that you mention the details in this section as well).

>> We appreciate for the helpful comment. Following the suggestion, we moved the description of CD8+ T cell culture in section 2.3 to section 2.7, thereby provide detailed procedure more clearly.

  1. The authors have mentioned in Figure 4C that they have significantly showed increase level of CD8 and IFN-y but it’s not reflected in their graph. Also, line 342-346 or in the discussion, the author should mention the significant of this specific experiment or why did they measure IFN-y secretion and correlate this to the function of PD-L1 in the tumor (inhibit the activity and toxic effects of CD8+T cells). Also, in Figure 4B:  the author should add what is the percentage of CD8+ T cells from the infiltrating T lymphocyte?

>> We agree that the data need to be explained in more clear manner. In the Figure 4C, the two representative histogram shows IFN-gamma/CD8 positive cell fraction in control (top) or siRNA nanoparticle-treated tumors (n=5 per each group). The number in the first quadrant, therefore, indicates IFN-gamma positive CD8 Cells. As marked in each histogram, it is increased from 4.53 to 17.23, implying the knockdown of PD-L1 by siRNA increases activated immune cell portion. The graph on right shows averaged, relative level of IFN-gamma, confirming the knockdown of PD-L1 indeed increases intratumoral IFN-gamma level. These data collectively suggest the knockdown of PD-L1 by siRNA successfully abrogates immune suppressive function of PDAC tumor and increase the activity of antitumoral immune response. We added this description in the result section 3.3

>> Regarding the data in Figure 4B, the percentage of CD8+ T cells from the infiltrating T lymphocyte is presented in the supplementary figure 4A. The fraction of hCD45+hCD19+ B cells were slightly increased from 5.6% to 8.0% which is consistent with the decrease of B cell composition in blood (Supplementary Figure 4B). We added this description in the result section 3.3

  1. Figure 5: Why did the authors opt to randomly select 6 grids for analysis? What happen if the authors select the whole area (tumor) and analyze based on cell numbers (in this case, immune cell infiltrates? Or could the authors focus on the area where there are more immune infiltrates and analyse from there the markers (CD45RO (Activated and memory T lymphocytes), CD4 and CD8 (T cell), FoxP3 (Treg cell), Granzyme (Activated NK and T cell), and for PD-L1 (whole tumor), as I understand there is tumor heterogeneity that’s why the results are not significant.

>> We appreciate for the critical comment. In fact, we selected 6 random grids as the whole section analysis is too time-consuming. Also, the whole-section analysis will include central part of tumor, that is often necrotic and rarely contains immune cells. To avoid any bias by selecting subregion, we randomly selected 6 regions on the peripheral region of each tumor.

  1. Mention how your results compare to other studies on RNAi-Mediated PD-L1 in Pancreatic Cancer Immunotherapy. 

>> Following the comment, we added recent reports about  RNAi mediated PD-L1 suppression in hepatocellular carcinoma and triple negative breast cancer.

  1. The authors should be consistent in labelling experimental groups (i.e. Figure 4 Ct/ Cont/ Control).

>> We thank for the helpful comment. We unified the control as Con.

  1. The authors should edit the following lines (for grammar):

Line 113-114

Line 263

Line 339: vesicle-treated mice or vehicle-treated mice

 >> We appreciate for the detailed point. It should be vehicle-treated, so we changed it.

Major:

  1. Methods/Results. The authors failed to detail the characteristics of siRNA-loaded PLGA NPs in terms of size, zeta potential. Also, they did not mention how did they account or measure encapsulation efficiency and how do they know if they are administering same amount to each mouse (siRNA loaded PLGA NPs)? Also, it is not clear whether the control group (in vivo) is PBS only or a scrambled siRNA -loaded PLGA. The authors should provide the sequences of siRNAs used

>>To show the particle size and zeta potential, we have added DLS analysis results in section 3.1 as follows,

Based on the dynamic light scattering analysis, the particle sizes of empty PLGA NPs and siRNA@PLGA NPs were 174.8 ± 2.4 and 188.5 ± 1.2 nm, respectively (Figure 1B). The negative charge in the empty PLGA NPs (-5.552 mV) became slightly neutralized in siRNA@PLGA NPs (-3.364 mV) after the positively charged PLL/siRNAs were complexed.”. Also, we have added 2.15 section in the Materials and Method as follows,

The dynamic diameter of zeta potential of empty PLGA NPs and siRNA@PLGA NPs were measured using a Malvern Nano ZS and Zeta-sizer (Malvern). Samples were serially diluted and each data were collected at a scattering angle of 173° with a 633 nm laser.”.

>> We described how to measure the amounts of siRNA in siRNA@PLGA NPs in section 2.1 as follows,

The siPD-L1 loading efficiency was measured using a Nanodrop spectrophotometer (Thermo Fisher Scientific), according to a previously proposed equation [19]. These measurements showed that 2 mg/mL of siRNA@PLGA NPs contained 0.3 mg/mL of siRNA.”. Therefore, given that concentration and volume of siRNA@PLGA NPs were known, the amount of siRNA administered to each mouse in the in vivo experiments could be calculated.

>> In the in vivo experiment of Figure 3B and 3C, the only PBS-treated control set indicates that OVA-nonspecific CD8+ T cells isolated from non-immunized C57BL/6 mice (i) were used as a control. To make it clear, we rewritten the sentence in section 3.2 as follows,

“According to the FI of the lysed cell contents, the siPD-L1@PLGA-treated sets (ii–iv) exhibited increased cytotoxicity of CTLs against Blue-OVA cells at both the 1:1 and 5:1 ratio, compared with the only PBS-treated control set without immunization (Figures 3B and 3C).”.

>> We have provided the sequences of siRNAs in section 2.4 as follows,

All oligoes were purchased from Bioneer Co. (Korea) and the sequences of siRNA were as follows: 5’-GCAGUGACCAUCAAGUCCUdTdT-3’ (human sense siPD-L1), 5’-dTdTAGGACUUGAUGGUCACUGC-3’ (human antisense siPD-L1), 5’- CCUACGCCACCAAUUUCGUdTdT-3’ (scrambled sense siRNA), 5’- dTdTGGAUGCGGUGGUUAAAGCA-3’ (scrambled antisense siRNA).”.

  1. Figure 1: It is unclear whether the authors treat the primary cells with scPD-L1 (served as control/untreated) for Figure 1B-1C or is it only stimulation with IFN then leave it for 68h without scrambled siRNA? If the authors didn’t, they should include an appropriate control such as the scrambled siRNA. For Figure 1D. Where is the cytotoxicity for siPD-L1?

>> In Figure 1C-1D of the revised manuscript, the untreated control represents the only PBS-treated cells after IFN-γ stimulation. When we compared the suppression effects of scrambled siRNA incapsulated in NPs (scPD-L1@PLGA), we did not observe any suppression of PD-L1 expression, at both the RNA and protein levels. To include these contents, we added sentences in section 3.1 as follows,

siRNA for PD-L1 incapsulated in NPs (siPD-L1@PLGA) efficiently suppressed the PD-L1 expression of the cell, at both the RNA (Figure 1C) and protein levels (Figure 1D), when compared to only PBS-treated control after IFN-γ stimulation. As expected, the scrambled siRNA nanoparticles (scPD-L1@PLGA) showed no suppression of PD-L1 expression at both RNA and protein levels, similar to the untreated control (data not shown).”

>> In Figure 1E of the revised manuscript, we examined cell viability of scrambled scPD-L1@PLGA, but not siPD-L1@PLGA. When we increased the concentration of scPD-L1@PLGA beyond 6 mg/mL, cell viability of 12 mg/mL was about 84%. Given that the non-cytotoxic concentration range is defined as greater than 90% of cell viability, these results indicate that the concentration ranges below 6 mg/mL do not induce any cytotoxic effect in Blue #96 cells. To include these contents, we added several sentences in section 3.1 as follows,

“Up to 6 mg/mL, no toxic effect of the scrambled scPD-L1@PLGA was observed (Figure 1E). When the concentration of scPD-L1@PLGA increased to 12 mg/mL, cell viability was about 84% (data not shown). Given that the non-cytotoxic concentration range is defined as greater than 90% of cell viability, these results indicate that the concentration ranges below 6 mg/mL do not induce any cytotoxic effect in Blue #96 cells. We selected 2 mg/mL as an optimized concentration for in vitro experiments.”

  1. For Figure 2D, where is the scRNA-PLGA group like in Figure 2C?

>> In the Figure 2D, we have added the downregulation effects of scPD-L1@PLGA on the IFN-γ induced PD-L1 expression. Also, we have added one sentence at the end of section 3.1 as follows,

As expected, the scrambled scPD-L1@PLGA showed no downregulation of IFN-γ induced PD-L1 expression.”.

  1. Figure 2B-2E: The authors should include group with scrambled siRNA and not PBS only as their appropriate control to have a direct comparison and rule out the effect of PD-L1 (knockdown) for enhanced cytotoxicity of CTLs.

>> First of all, we assumed Figure 2B-2E in the reviewer’ comments are pointing the data in  3B-3E. Please correct us if it is wrong.

When we examined the cytotoxicity and proliferation of CTLs in the co-culture system with scrambled siPD-L1@PLGA-treated Blue-OVA cells, we observed the similar effects to PBS-treated Blue-OVA cells. To describe these effects of scrambled siRNA in the Figure 3B-3E, we have added two sentences in section 3.2 as follows,
As expected, the scrambled siPD-L1@PLGA-treated sets did not show an increase in the cytotoxicity of CTLs against Blue-OVA cells at both ratios, similar to the PBS-treated sets (data not shown).”, and
“As expected, the scrambled siPD-L1@PLGA-treated Blue-OVA cells did not show increased proliferation of CTLs at three different E:T ratios, similar to the untreated control set (data not shown).”.

  1. It is unclear whether the Ct group in Figure 4 is PBS treated mice or includes scrambled siRNA/PLGA as control. Also, mention how did the authors administer (route of administration) the siPD-L1@PLGA to the animals? And how much of siPD-L1 and PLGA NPs? When did they start their treatment?

>> We appreciate for the valuable comments and regret the procedure was not described in methods section. Thus, we edited section 2.11 and described details of the nanoparticle injection The Ct group received PBS so we edited Figure 4 legends as well.

Finally, the authors should add more animal numbers for their humanized PDX model; check the expression of PD-L1 in their patient derived cell cultures (PDX that they have used to inoculate to humanized NSG mice).

>> We thanks for the critical point. Unfortunately, we could not increase the number of mouse due to the high cost of the humanized NSG (from Jackson Laboratory). We agree, though, increasing the number will certainly increase the significance of the study, which we will pursue in the further study. For the PD-L1 expression of the PDAC PDC and PDX, we added the expression data in the Supplementary Figure 1B. We described this in the result section 3.3.

Discussion: Perhaps, they need to include in the discussion the history of the patient/s where they derived the PDX lines (any prior drug treatment, resistance to any treatments, PD-L1 status) to correlate with their findings. Since the humanized NSG model is not the best model to use for this study, perhaps they can do proof of concept experiments by growing these PDX lines (PDAC) in different matrices or in 3D co-cultured with CAFs, stellate cells or immune cells (PBMC) and test their siRNA-PLGA in combination with another therapy/drug.

>> We deeply appreciate for the valuable suggestion for this subject and agree with the other model such as different matrices or in 3D co-cultured with CAFs, stellate cells or immune cells (PBMC). We will design such system and apply our nanoparticle in the further study. For the patient information for the PDX, please check this reference (Ref.#26 in this manuscript)

  1. Jung, J.; Lee, C.H.; Seol, H.S.; Choi, Y.S.; Kim, E.; Lee, E.J.; Rhee, J.K.; Singh, S.R.; Jun, E.S.; Han, B., et al. Generation and molecular characterization of pancreatic cancer patient-derived xenografts reveals their heterologous nature. Oncotarget 2016, 7, 62533-62546, doi:10.18632/oncotarget.11530.

We added this point in the method section 2.2.

Round 2

Reviewer 1 Report

The authors sufficently addressed my points. The manuscript has improved and is now suitable for publication.

Reviewer 2 Report

The authors have adequately addressed my comments raised in a previous round of review. Just minor spell check required.